# POMO: Policy Optimization with Multiple Optima for Reinforcement Learning

**Yeong-Dae Kwon, Jinho Choo, Byoungjip Kim, Iljoo Yoon, Youngjune Gwon, Seungjai Min**
Samsung SDS
{y.d.kwon, jinho12.choo, bjip.kim, iljoo.yoon, gyj.gwon, seungjai.min}@samsung.com

## Abstract

In neural combinatorial optimization (CO), reinforcement learning (RL) can turn a deep neural net into a fast, powerful heuristic solver of NP-hard problems. This approach has a great potential in practical applications because it allows near-optimal solutions to be found without expert guides armed with substantial domain knowledge. We introduce Policy Optimization with Multiple Optima (POMO), an end-to-end approach for building such a heuristic solver. POMO is applicable to a wide range of CO problems. It is designed to exploit the symmetries in the representation of a CO solution. POMO uses a modified REINFORCE algorithm that forces diverse rollouts towards all optimal solutions. Empirically, the low-variance baseline of POMO makes RL training fast and stable, and it is more resistant to local minima compared to previous approaches. We also introduce a new augmentation-based inference method, which accompanies POMO nicely. We demonstrate the effectiveness of POMO by solving three popular NP-hard problems, namely, traveling salesman (TSP), capacitated vehicle routing (CVRP), and 0-1 knapsack (KP). For all three, our solver based on POMO shows a significant improvement in performance over all recent learned heuristics. In particular, we achieve the optimality gap of 0.14% with TSP100 while reducing inference time by more than an order of magnitude.

## 1   Introduction

Combinatorial optimization (CO) is an important problem in logistics, manufacturing and distribution supply chain, and sequential resource allocation. The problem is studied extensively by Operations Research (OR) community, but the real-world CO problems are ubiquitous, and each problem is different from one another with its unique constrains. Moreover, these constrains tend to vary rapidly with a changing work environment. Devising a powerful and efficient algorithm that can be applied uniformly under various conditions is tricky, if not impossible. Therefore, many CO problems faced in industries have been commonly dealt with hand-crafted heuristics, despite their drawbacks, engineered by local experts.

In the field of computer vision (CV) and natural language processing (NLP), classical methods based on manual feature engineering by experts have now been superseded by automated end-to-end deep learning algorithms [1, 2, 3, 4, 5]. Tremendous progresses in supervised learning, where a mapping from training inputs to their labels is learned, has made this remarkable transition possible. Unfortunately, supervised learning is largely unfit for most CO problems because one cannot have an instant access to optimal labels. Rather, one should make use of the scores, that are easily calculable for most CO solutions, to train a model. Reinforcement learning paradigm suits combinatorial optimization problems very well.

Recent approaches in deep reinforcement learning (RL) have been promising [6], finding close-to-optimal solutions to the abstracted NP-hard CO problems such as traveling salesman (TSP) [7, 8, 9, 10,

11, 12], capacitated vehicle routing (CVRP) [10, 11, 13, 14, 15], and 0-1 knapsack (KP) [7] in superior speed. We contribute to this line of group effort in the deep learning community by introducing Policy Optimization with Multiple Optima (POMO). POMO offers a simple and straightforward framework that can automatically generate a decent solver. It can be applied to a wide range of general CO problems because it uses symmetry in the CO itself, found in sequential representation of CO solutions.

We demonstrate the effectiveness of POMO by solving three NP-hard problems aforementioned, namely TSP, CVRP, and KP, using the same neural net and the same training method. Our approach is purely data-driven, and the human guidance in the design of the training procedure is kept to minimal. More specifically, it does not require problem-specific hand-crafted heuristics to be inserted into its algorithms. Despite its simplicity, our experiments confirm that POMO achieves superior performances in reducing the optimality gap and inference time against all contemporary neural RL approaches.

The contribution of this paper is three-fold:

- We identify symmetries in RL methods for solving CO problems that lead to multiple optima. Such symmetries can be leveraged during neural net training via parallel multiple rollouts, each trajectory having a different optimal solution as its goal for exploration.

- We devise a new low-variance baseline for policy gradient. Because this baseline is derived from a group of heterogeneous trajectories, learning becomes less vulnerable to local minima.

- We present the inference method based on multiple greedy rollouts that is more effective than the conventional sampling inference method. We also introduce an instance augmentation technique that can further exploit symmetries of CO problems at the inference stage.

## 2 Related work

**Deep RL construction methods.** Bello *et al.* [7] use a Pointer Network (PtrNet) [16] in their neural combinatorial optimization framework. As one of the earliest deep RL approaches, they have employed the actor-critic algorithm [17] and demonstrated neural combinatorial optimization that achieves close-to-optimal results in TSP and KP. The PtrNet model is based on the sequence-to-sequence architecture [3] and uses attention mechanism [18]. Narari *et al.* [19] have further improved the PtrNet model.

Differentiated from the previous recurrent neural network (RNN) based approaches, Attention Model [10] opts for the Transformer [4] architecture. REINFORCE [20] with a greedy rollout baseline trains Attention Model, similar to self-critical training [21]. Attention Model has been applied to routing problems including TSP, orienteering (OP), and VRP. Peng *et al.* [22] show that a dynamic use of Attention Model can enhance its performance.

Dai *et al.* propose Struct2Vec [23]. Using Struct2Vec, Khalil *et al.* [8] have developed a deep Q-learning [24] method to solve TSP, minimum vertex cut and maximum cut problems. Partial solutions are embedded as graphs, and the deep neural net estimates the value of each graph.

**Inference techniques.** Once the neural net is fully trained, inference techniques can be used to improve the quality of solutions it produces. Active search [7] optimizes the policy on a single test instance. Sampling method [7, 10] is used to choose the best among the multiple solution candidates. Beam search [16, 9] uses advanced strategies to improve the efficiency of sampling. Classical heuristic operations as post-processing may also be applied on the solutions produced by the neural net [25, 22] to further enhance their quality.

**Deep RL improvement methods.** POMO belongs to the category of *construction* type RL method summarized above, in which a CO solution is created by the neural net in one shot. There is, however, another important class of RL approach for solving CO problems that combines machine learning with the existing heuristic methods. A neural net can be trained to guide local search algorithm, which iteratively finds a better solution based on the previous ones until the time budget runs out. Such *improvement* type RL methods have been demonstrated with outstanding results by many, including Wu *et al.* [11] and Costa *et al.* [12] for TSP and Chen & Tian [13], Hottung & Tierney [14],

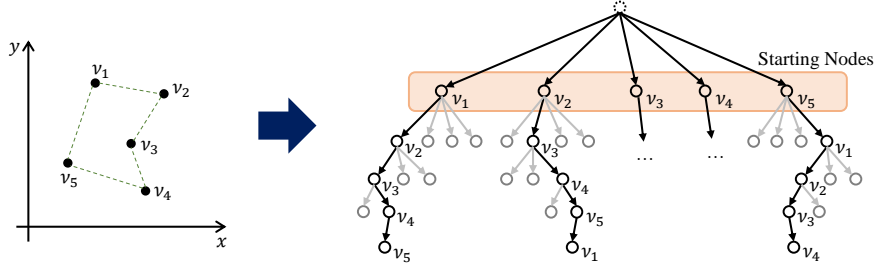

Figure 1: Multiple optimal solutions of TSP highlighted in tree search. For the given instance of 5-node TSP problem, there exists only one unique optimal solution (LEFT). But when this solution is represented as a sequence of nodes, multiple representations exist (RIGHT).

Lu *et al.* [15] for CVRP. We note that formulation of improvement heuristics on top of POMO should be possible and can be an important further research topic.

## 3 Motivation

Assume we are given a combinatorial optimization problem with a group of nodes $\{v_1, v_2, \ldots, v_M\}$ and have a trainable policy net parameterized by $\theta$ that can produce a valid solution to the problem. A solution $\boldsymbol{\tau} = (a_1, \ldots, a_M)$, where the $i$th action $a_i$ can choose a node $v_j$, is generated by the neural net autoregressively one node at a time, following the stochastic policy

$$\pi_t = \begin{cases} p_\theta(a_t \mid s) & \text{for} \quad t = 1 \\ p_\theta(a_t \mid s, a_{1:t-1}) & \text{for} \quad t \in \{2, 3, \ldots, M\} \end{cases} \tag{1}$$

where $s$ is the state defined by the problem instance.

In many cases, a solution of a CO problem can take multiple forms when represented as a sequence of nodes. A routing problem that contains a loop, or a CO problem finding a "set" of items have such characteristics, to name a few. Take TSP for an example: if $\boldsymbol{\tau} = (v_1, v_2, v_3, v_4, v_5)$ is an optimal solution of a 5-node TSP, then $\boldsymbol{\tau}' = (v_2, v_3, v_4, v_5, v_1)$ also represents the same optimal solution (Figure 1).

When asked to produce the best possible answer within its capability, a perfectly logical agent with prior knowledge of equivalence among such sequences should produce the same solution regardless of which node it chooses to output first. This, however, has not been the case in the previous learning-based models. As is clear in Equation (1), the starting action $(a_1)$ heavily influences the rest of the agent's course of actions $(a_2, a_3, \ldots, a_M)$, when in fact any choice for $a_1$ should be equally good[1]. We seek to find a policy optimization method that can fully exploit this symmetry.

## 4 Policy Optimization with Multiple Optima (POMO)

### 4.1 Explorations from multiple starting nodes

POMO begins with designating $N$ different nodes $\{a_1^1, a_1^2, \ldots, a_1^N\}$ as starting points for exploration (Figure 2, (b)). The network samples $N$ solution trajectories $\{\boldsymbol{\tau}^1, \boldsymbol{\tau}^2, \ldots, \boldsymbol{\tau}^N\}$ via Monte-Carlo method, where each trajectory is defined as a sequence

$$\boldsymbol{\tau}^i = (a_1^i, a_2^i, \ldots, a_M^i) \qquad \text{for} \quad i = 1, 2, \ldots, N. \tag{2}$$

In previous work that uses RNN- or Transformer-style neural architectures, the first node from multiple sample trajectories is always chosen by the network. A trainable START token, a legacy from NLP where those models originate, is fed into the network, and the first node is returned (Figure 2, (a)). Normally, the use of such START token is sensible, because it allows the machine

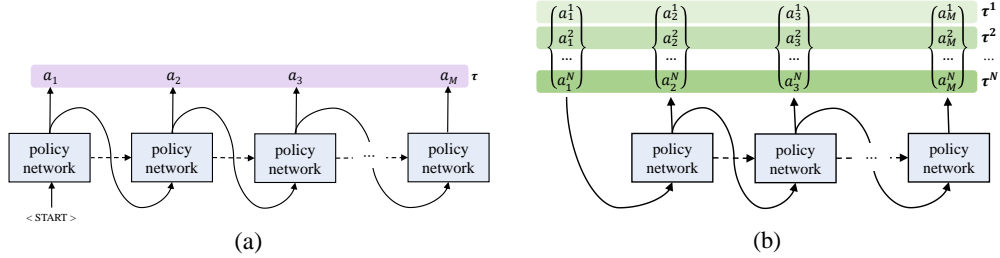

(a)                                                    (b)

Figure 2: (a) Common method for generating a single solution trajectory ($\boldsymbol{\tau}$) based on START token scheme. (b) POMO method for multiple trajectory $\{\boldsymbol{\tau}^1, \boldsymbol{\tau}^2, ..., \boldsymbol{\tau}^N\}$ generation in parallel with a different starting node for each trajectory.

to learn to find the "correct" first move that leads to the best solution. In the presence of multiple "correct" first moves, however, it forces the machine to favor particular starting points, which may lead to a biased strategy.

When all first moves are equally good, therefore, it is wise to apply entropy maximization techniques [27] to improve exploration. Entropy maximization is typically carried out by adding an entropy regularization term to the objective function of RL. POMO, however, directly maximizes the entropy on the first action by forcing the network to always produce multiple trajectories, all of them contributing equally during training.

Note that these trajectories are fundamentally different from repeatedly sampled $N$ trajectories under the START token scheme [28]. Each trajectory originating from a START token stays close to a single optimal path, but $N$ solution trajectories of POMO will closely match $N$ different node-sequence representations of the optimal solution. Conceptually, explorations by POMO are analogous to guiding a student to solve the same problem repeatedly from many different angles, exposing her to a variety of problem-solving techniques that would otherwise be unused.

## 4.2 A shared baseline for policy gradients

POMO is based on the REINFORCE algorithm [20]. Once we sample a set of solution trajectories $\{\boldsymbol{\tau}^1, \boldsymbol{\tau}^2, \dots, \boldsymbol{\tau}^N\}$, we can calculate the return (or total reward) $R(\boldsymbol{\tau}^i)$ of each solution $\boldsymbol{\tau}^i$. To maximize the expected return $J$, we use gradient ascent with an approximation

$$\nabla_\theta J(\theta) \approx \frac{1}{N} \sum_{i=1}^{N} (R(\boldsymbol{\tau}^i) - b^i(s)) \nabla_\theta \log p_\theta(\boldsymbol{\tau}^i|s) \tag{3}$$

where $p_\theta(\boldsymbol{\tau}^i|s) \equiv \prod_{t=2}^{M} p_\theta(a_t^i|s, a_{1:t-1}^i)$.

In Equation (3), $b^i(s)$ is a baseline that one has some freedom of choice to reduce the variance of the sampled gradients. In principle, it can be a function of $a_1^i$, assigned differently for each trajectory $\boldsymbol{\tau}^i$. In POMO, however, we use the shared baseline,

$$b^i(s) = b_{\text{shared}}(s) = \frac{1}{N} \sum_{j=1}^{N} R(\boldsymbol{\tau}^j) \quad \text{for all } i. \tag{4}$$

Algorithm 1 presents the POMO training with mini-batch.

POMO baseline induces less variance in the policy gradients compared to the greedy-rollout baseline [10]. The advantage term in Equation (3), $R(\boldsymbol{\tau}^i) - b^i(s)$, has zero mean for POMO, whereas the greedy-rollout baseline results in negative advantages most of the time. This is because sample-rollouts (following `softmax` of the policy) have difficulty in surpassing greedy-rollouts (following `argmax` of the policy) in terms of the solution qualities, as will be demonstrated later in this paper. Also, as an added benefit, POMO baseline can be computed efficiently compared to other baselines used in previous deep-RL construction methods, which require forward passes through either a separately trained network (Critic [7, 8]) or the cloned policy network (greedy-rollout [10]).

Most importantly, however, the shared baseline used by POMO makes RL training highly resistant to local minima. After generating $N$ solution trajectories $\{\boldsymbol{\tau}^1, \boldsymbol{\tau}^2, \dots, \boldsymbol{\tau}^N\}$, if we do not use the shared

---
**Algorithm 1** POMO Training
---
1: **procedure** TRAINING(training set $S$, number of starting nodes per sample $N$, number of training steps $T$, batch size $B$)
2:      initialize policy network parameter $\theta$
3:      **for** $step = 1, \ldots, T$ **do**
4:          $s_i \leftarrow \text{SAMPLEINPUT}(S) \quad \forall i \in \{1, \ldots, B\}$
5:          $\{\alpha_i^1, \alpha_i^2, \ldots, \alpha_i^N\} \leftarrow \text{SELECTSTARTNODES}(s_i) \quad \forall i \in \{1, \ldots, B\}$
6:          $\boldsymbol{\tau}_i^j \leftarrow \text{SAMPLEROLLOUT}(\alpha_i^j, s_i, \pi_\theta) \quad \forall i \in \{1, \ldots, B\}, \forall j \in \{1, \ldots, N\}$
7:          $b_i \leftarrow \frac{1}{N} \sum_{j=1}^{N} R(\boldsymbol{\tau}_i^j) \quad \forall i \in \{1, \ldots, B\}$
8:          $\nabla_\theta J(\theta) \leftarrow \frac{1}{BN} \sum_{i=1}^{B} \sum_{j=1}^{N} (R(\boldsymbol{\tau}_i^j) - b_i) \nabla_\theta \log p_\theta(\boldsymbol{\tau}_i^j)$
9:          $\theta \leftarrow \theta + \alpha \nabla_\theta J(\theta)$
10:     **end for**
11: **end procedure**
---

baseline but strictly stick to the greedy-rollout baseline scheme [10] instead, each sample-rollout $\boldsymbol{\tau}^i$ would be assessed independently. Actions that produced $\boldsymbol{\tau}^i$ would be reinforced solely by how much better (or worse) it performed compared to its greedy-rollout counterpart with the same starting node $a_1^i$. Because this training method is guided by the difference between the two rollouts produced by two closely-related networks, it is likely to converge prematurely at a state where both the actor and the critic underperform in a similar fashion. With the shared baseline, however, each trajectory now competes with $N - 1$ other trajectories where no two trajectories can be identical. With the increased number of heterogeneous trajectories all contributing to setting the baseline at the right level, premature converge to a suboptimal policy is heavily discouraged.

### 4.3 Multiple greedy trajectories for inference

Construction type neural net models for CO problems have two modes for inference in general. In "greedy mode," a single deterministic trajectory is drawn using `argmax` on the policy. In "sampling mode," multiple trajectories are sampled from the network following the probabilistic policy. Sampled solutions may return smaller rewards than the greedy one on average, but sampling can be repeated as many times as needed at the computational cost. With a large number of sampled solutions, some solutions with rewards greater than that of the greedy rollout can be found.

Using the multi-starting-node approach of POMO, however, one can produce not just one but multiple greedy trajectories. Starting from $N$ different nodes $\{a_1^1, a_1^2, \ldots, a_1^N\}$, $N$ different greedy trajectories can be acquired deterministically, from which one can choose the best similarly to the "sampling mode" approach. $N$ greedy trajectories are in most cases superior than $N$ sampled trajectories.

**Instance augmentation.** One drawback of POMO's multi-greedy inference method is that $N$, the number of greedy rollouts one can utilize, cannot be arbitrarily large, as it is limited to a finite number of possible starting nodes. In certain types of CO problems, however, one can bypass this limit by introducing *instance augmentation*. It is a natural extension from the core idea of POMO, seeking different ways to arrive at the same optimal solution. What if you can reformulate the problem, so that the machine sees a different problem only to arrive at the exact same solution? For example, one can flip or rotate the coordinates of all the nodes in a 2D routing optimization problem and generate another instance, from which more greedy trajectories can be acquired.

Instance augmentation is inspired by self-supervised learning techniques that train neural nets to learn the equivalence between rotated images [29]. For CV tasks, there are conceptually similar test-time augmentation techniques such as "multi-crop evaluation" [30] that enhance neural nets' performance at the inference stage. Applicability and multiplicative power of instance augmentation technique on CO tasks depend on the specifics of a problem and also on the policy network model that one uses. More ideas on instance augmentation are described in Appendix.

Algorithm 2 describes POMO's inference method, including the instance augmentation.

**Algorithm 2** POMO Inference
---
1: **procedure** INFERENCE(input $s$, policy $\pi_\theta$, number of starting nodes $N$, number of transforms $K$)
2:      $\{s_1, s_2, \ldots, s_K\} \leftarrow$ AUGMENT($s$)
3:      $\{\alpha_k^1, \alpha_k^2, \ldots, \alpha_k^N\} \leftarrow$ SELECTSTARTNODES($s_k$)    $\forall k \in \{1, \ldots, K\}$
4:      $\boldsymbol{\tau}_k^j \leftarrow$ GREEDYROLLOUT($\alpha_k^j, s, \pi_\theta$)    $\forall j \in \{1, \ldots, N\}, \forall k \in \{1, \ldots, K\}$
5:      $k_{\max}, j_{\max} \leftarrow \mathrm{argmax}_{k,j} \, R(\boldsymbol{\tau}_k^j)$
6:      **return** $\boldsymbol{\tau}_{k_{\max}}^{j_{\max}}$
7: **end procedure**
---

## 5 Experiments

**The Attention Model.** All of our POMO experiments use the policy network named Attention Model (AM), introduced by Kool *et al.* [10]. The AM is particularly well-suited for POMO, although we emphasize that POMO is a general RL method, not tied to a specific structure of the policy network. The AM consists of two main components, an encoder and a decoder. The majority of computation happens inside the heavy, multi-layered encoder, through which information of each node and its relation with other nodes is embedded as a vector. The decoder then generates a solution sequence autoregressively using these vectors as the keys for its dot-product attention mechanism.

To apply POMO, we need to draw multiple ($N$) trajectories for one instance of a CO problem. This does not affect the encoding procedure on the AM, as the encoding is required only once regardless of the number of trajectories one needs to generate. The decoder of the AM, on the other hand, needs to process $N$ times more computations for POMO. By stacking $N$ queries into a single matrix and passing it to the attention mechanism (a natural usage of attention), $N$ trajectories can be generated efficiently in parallel.

**Problem setup.** For TSP and CVRP, we solve the problems with a setup as prescribed in Kool *et al.* [10]. For 0-1 KP, we follow the setup by Bello *et al.* [7].

**Training.** For all experiments, policy gradients are averaged from a batch of 64 instances. Adam optimizer [31] is used with a learning rate $\eta = 10^{-4}$ and a weight decay ($L_2$ regularization) $w = 10^{-6}$. To keep the training condition simple and identical for all experiments we have not applied a decaying learning rate, although we recommend a fine-tuned decaying learning rate in practice for faster convergence. We define one epoch 100,000 training instances generated randomly on the fly. Training time varies with the size of the problem, from a couple of hours to a week. In the case of TSP100, for example, one training epoch takes about 7 minutes on a single Titan RTX GPU. We have waited as long as 2,000 epochs ($\sim$ 1 week) to observe full converge, but as shown in Figure 3 (b), most of the learning is already completed by 200 epochs ($\sim$ 1 day).

**Inference.** We follow the convention and report the time for solving 10,000 random instances of each problem. For routing problems, we have performed inferences with and without $\times 8$ instance augmentation, using the coordinate transformations listed in Table 1. No instance augmentation is used for KP because there is no straightforward way to do so.

Table 1: Unit square transformations

| $f(x,y)$ | |
| --- | --- |
| $(x, y)$ | $(y, x)$ |
| $(x, 1\text{-}y)$ | $(y, 1\text{-}x)$ |
| $(1\text{-}x, y)$ | $(1\text{-}y, x)$ |
| $(1\text{-}x, 1\text{-}y)$ | $(1\text{-}y, 1\text{-}x)$ |

Originally, Kool *et al.* [10] have trained the AM using REINFORCE with a greedy rollout baseline. It is interesting to see how much the performance improves when POMO is used for training instead. For a concrete ablation study, however, the two separately trained neural nets must be evaluated in the same way. Because POMO inference method chooses the best from multiple answers, even without the instance augmentation, this gives POMO an unfair advantage. Therefore, we have additionally performed the inference in "single-trajectory mode" on our POMO-trained network, in which a random starting node is chosen to draw a single greedy rollout.

Note that averaged inference results can fluctuate when they are based on a small (10,000) test set of random instances. In order to avoid confusion for the readers, we have slightly modified some of the averaged path lengths of TSP results in Table 2 (based on the reported optimality gaps) so that they

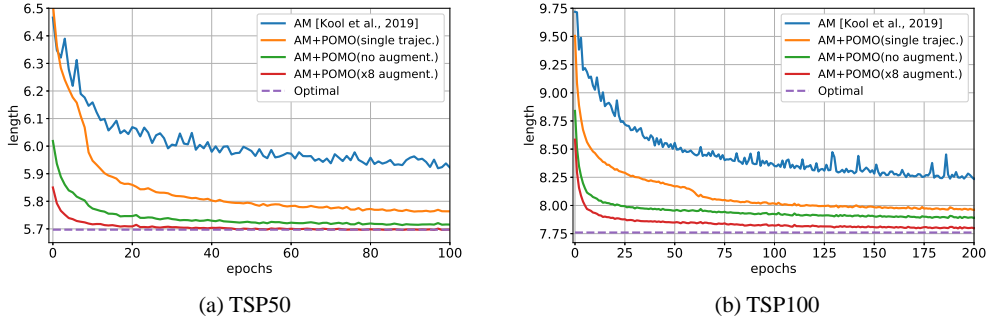

|                     | (a) TSP50 | (b) TSP100 |
|---------------------|-----------|------------|

Figure 3: Learning curves for TSP50 and TSP100 of REINFORCE with a greedy rollout baseline [10] (blue) and those of POMO (orange, green, and red) made by three different inference methods on the same neural net (AM). After each training epoch, we generate 10,000 random instances on the fly and use them as a validation set.

are consistent with the optimal values we computed (using more than 100,000 samplings), 3.83 and 5.69 for TSP20 and TSP50, respectively. For CVRP and KP, there are even larger sampling errors than TSP, and thus we are more careful in the presentation of the results in this case. We display "gaps" in the result tables only when they are based on the same test sets.

**Code.** Our implementation of POMO on the AM using PyTorch is publicly available[2]. We also share a trained model for each problem and its evaluation code.

## 5.1 Traveling salesman problem

We implement POMO by assigning every node to be a starting point for a sample rollout for TSP. That is, the number of starting nodes ($N$) we use is 20 for TSP20, 50 for TSP50, and 100 for TSP100.

In Table 2 we compare the performance of POMO on TSP with other baselines. The first group of baselines shown at the top are results from Concorde [32] and a few other representative non-learning-based heuristics. We have run Concorde ourselves to get the optimal solutions, and other solvers' data are adopted from Wu *et al.* [11] and Kool *et al.* [10]. The second group of baselines are from deep RL improvement-type approaches found in the literature [9, 11, 12]. In the third group, we present

Table 2: Experiment results on TSP

| Method | TSP20 | | | TSP50 | | | TSP100 | | |
|--------|-------|-----|------|-------|-----|------|--------|-----|------|
|        | Len.  | Gap | Time | Len.  | Gap | Time | Len.   | Gap | Time |
| Concorde          | 3.83 | -      | (5m)   | 5.69 | -      | (13m)  | 7.76 | -      | (1h)  |
| LKH3              | 3.83 | 0.00%  | (42s)  | 5.69 | 0.00%  | (6m)   | 7.76 | 0.00%  | (25m) |
| Gurobi            | 3.83 | 0.00%  | (7s)   | 5.69 | 0.00%  | (2m)   | 7.76 | 0.00%  | (17m) |
| OR Tools          | 3.86 | 0.94%  | (1m)   | 5.85 | 2.87%  | (5m)   | 8.06 | 3.86%  | (23m) |
| Farthest Insertion | 3.92 | 2.36% | (1s)   | 6.00 | 5.53%  | (2s)   | 8.35 | 7.59%  | (7s)  |
| GCN [9], beam search | 3.83 | 0.01% | (12m) | 5.69 | 0.01% | (18m) | 7.87 | 1.39% | (40m) |
| Improv. [11], {5000} | 3.83 | 0.00% | (1h)  | 5.70 | 0.20% | (1h)  | 7.87 | 1.42% | (2h)  |
| Improv. [12], {2000} | 3.83 | 0.00% | (15m) | 5.70 | 0.12% | (29m) | 7.83 | 0.87% | (41m) |
| AM [10], greedy   | 3.84 | 0.19% | ($\ll$1s) | 5.76 | 1.21% | (1s) | 8.03 | 3.51% | (2s)  |
| AM [10], sampling | 3.83 | 0.07% | (1m)      | 5.71 | 0.39% | (5m) | 7.92 | 1.98% | (22m) |
| POMO, single trajec. | 3.83 | 0.12% | ($\ll$1s) | 5.73 | 0.64% | (1s)  | 7.84 | 1.07% | (2s)  |
| POMO, no augment.    | 3.83 | 0.04% | ($\ll$1s) | 5.70 | 0.21% | (2s)  | 7.80 | 0.46% | (11s) |
| POMO, $\times$8 augment. | 3.83 | 0.00% | (3s)      | 5.69 | 0.03% | (16s) | 7.77 | 0.14% | (1m)  |

the results from the AM that is trained by our implementation of REINFORCE with a greedy rollout baseline [10] instead of POMO.

Given 10,000 random instances of TSP20 and TSP50, POMO finds near-optimal solutions with optimality gaps of 0.0006% in seconds and 0.025% in tens of seconds, respectively. For TSP100, POMO achieves the optimality gap of 0.14% in a minute, outperforming all other learning-based heuristics significantly, both in terms of the quality of the solutions and the time it takes to solve.

In the table, results under "AM, greedy" method and "POMO, single trajec." method are both from the identical network structure that is tested by the same inference technique. The only difference was training, so the substantial improvement (*e.g.* from 3.51% to 1.07% in optimality gap on TSP100) indicates superiority of the POMO training method. As for the inference techniques, it is shown that the combined use of multiple greedy rollouts of POMO and the $\times 8$ instance augmentation can reduce the optimality gap even further, by an order of magnitude.

Learning curves of TSP50 and TSP100 in Figure 3 show that POMO training is more stable and sample-efficient. In reading these graphs, one should keep in mind that POMO uses $N$-times more trajectories than simple REINFORCE for each training epoch. POMO training time is, however, comparable to that of REINFORCE, thanks to the parallel processing on trajectory generation. For example, TSP100 training takes about 7 minutes per epoch for POMO while it take 6 minutes for REINFORCE.

## 5.2 Capacitated vehicle routing problem

When POMO trains a policy net, ideally it should use only the "good" starting nodes from which one can roll out optimal solutions. But, unlike TSP, not all nodes in CVRP can be the first steps for optimal trajectories (see Figure 4) and there is no way of figuring out which of the nodes are good without actually knowing the optimal solution a priori. One way to resolve this issue is to add a secondary network that returns candidates for optimal starting nodes to be used by POMO. We leave this approach for future research, however, and in our CVRP experiment we stick to the same policy net that we have used for TSP without an upgrade. We simply use all nodes as starting nodes for POMO exploration regardless of whether they are good or bad.

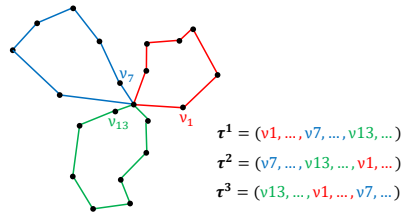

$$\tau^1 = (v1, \dots, v7, \dots, v13, \dots)$$
$$\tau^2 = (v7, \dots, v13, \dots, v1, \dots)$$
$$\tau^3 = (v13, \dots, v1, \dots, v7, \dots)$$

Figure 4: An optimal solution of a 20-node CVRP plotted as a graph. For an agent that makes selections in the counter-clock-wise direction[3], there are only three sequence representations of the optimal solution available: $\tau^1$, $\tau^2$, and $\tau^3$.

This naive way of applying POMO can still make a powerful solver. Experiment results on CVRP with 20, 50, and 100 customer nodes are reported in Table 3, and POMO is shown to outperform simple REINFORCE by a large margin. Note that there is no algorithm yet that can find optimal solutions of 10,000 random

Table 3: Experiment results on CVRP

| Method | CVRP20 | | | CVRP50 | | | CVRP100 | | |
|---|---|---|---|---|---|---|---|---|---|
| | Len. | Gap | Time | Len. | Gap | Time | Len. | Gap | Time |
| LKH3 | 6.12 | - | (2h) | 10.38 | - | (7h) | 15.68 | - | (12h) |
| OR Tools | 6.42 | 4.84% | (2m) | 11.22 | 8.12% | (12m) | 17.14 | 9.34% | (1h) |
| NeuRewriter [13] | 6.16 | | (22m) | 10.51 | | (18m) | 16.10 | | (1h) |
| NLNS [14] | 6.19 | | (7m) | 10.54 | | (24m) | 15.99 | | (1h) |
| L2I [15] | 6.12 | | (12m) | 10.35 | | (17m) | 15.57 | | (24m) |
| AM [10], greedy | 6.40 | 4.45% | ($\ll$1s) | 10.93 | 5.34% | (1s) | 16.73 | 6.72% | (3s) |
| AM [10], sampling | 6.24 | 1.97% | (3m) | 10.59 | 2.11% | (7m) | 16.16 | 3.09% | (30m) |
| POMO, single trajec. | 6.35 | 3.72% | ($\ll$1s) | 10.74 | 3.52% | (1s) | 16.15 | 3.00% | (3s) |
| POMO, no augment. | 6.17 | 0.82% | (1s) | 10.49 | 1.14% | (4s) | 15.83 | 0.98% | (19s) |
| POMO, $\times 8$ augment. | 6.14 | 0.21% | (5s) | 10.42 | 0.45% | (26s) | 15.73 | 0.32% | (2m) |

[3]Empirically, we find that neural net-based agents choose nodes in an orderly fashion.

Table 4: Experiment results on KP

| Method | KP50 | | KP100 | | KP200 | |
|---|---|---|---|---|---|---|
| | Score | Gap | Score | Gap | Score | Gap |
| Optimal | 20.127 | - | 40.460 | - | 57.605 | - |
| Greedy Heuristics | 19.917 | 0.210 | 40.225 | 0.235 | 57.267 | 0.338 |
| Pointer Net [7], greedy | 19.914 | 0.213 | 40.217 | 0.243 | 57.271 | 0.334 |
| AM [10], greedy | 19.957 | 0.173 | 40.249 | 0.211 | 57.280 | 0.325 |
| POMO, single trajec. | 19.997 | 0.130 | 40.335 | 0.125 | 57.345 | 0.260 |
| POMO, no augment. | 20.120 | 0.007 | 40.454 | 0.006 | 57.597 | 0.008 |

CVRP instances in a reasonable time, so the "Gap" values in the table are given relative to LKH3 [33] results. POMO has a smaller gap in CVRP100 (0.32%) than in CVRP50 (0.45%), which is probably due to LKH3 falling faster in performance than POMO as the size of the problem grows.

In fact, L2I recently developed by Lu *et al.* [15] performs better than LKH3, making it the first deep RL-based CVRP solver that beats classical non-learning-based OR methods. Their achievement is a milestone in the application of the deep learning to OR. To emphasize the differences between POMO and L2I (other than the speed), POMO is a general RL tool that can be applied to many different CO problems in a purely data-driven way. One the other hand, L2I is a specialized routing problem solver based on a handcrafted pool of improvement operators. Because POMO is a construction method and L2I is an improvement type, it is possible to combine the two methods to produce even better results.

### 5.3  0-1 knapsack problem

We choose KP to demonstrate flexibility of POMO beyond routing problems. Similarly to the case of CVRP, we reuse the neural net for TSP and take the naive approach that uses all items given in the instance as the first steps for rollouts, avoiding an additional, more sophisticated "SelectStartNodes" network to be devised. In solving KP, a weight and a value of each item replaces x- and y-coordinate of each node of TSP. As the network generates a sequence of items, we put these items into the knapsack one by one until the bag is full, at which point we terminate the sequence generation.

In Table 4, the POMO results are compared with the optimal solutions based on dynamic programming, as well as those by greedy heuristics and our implementation of PtrNet [7] and the original AM method [10]. Even without the instance augmentation, POMO greatly improves the quality of the solutions one can acquire from a deep neural net.

## 6  Conclusion

POMO is a purely data-driven combinatorial optimization approach based on deep reinforcement learning, which avoids hand-crafted heuristics built by domain experts. POMO leverages the existence of multiple optimal solutions of a CO problem to efficiently guide itself towards the optimum, during both the training and the inference stages. We have empirically evaluated POMO with traveling salesman (TSP), capacitated vehicle routing (CVRP), and 0-1 knapsack (KP) problems. For all three problems, we find that POMO achieves the state-of-the-art performances in closing the optimality gap and reducing the inference time over other construction-type deep RL methods.

## Broader Impact

This work can facilitate the use of reinforcement learning approach based on deep neural net as a replacement for traditional heuristic methods used in operations of many sectors of business. Better and easier-to-use optimization tools will increase the productivity, but may lead to automation of works that previously needed more manual operations (jobs).

## Acknowledgments and Disclosure of Funding

We thank anonymous reviewers for their comments, as they have helped improving the paper very much. We declare no third party funding or support.

## Footnotes

[1]A cognitive bias found in psychology called "anchoring" [26] is the human equivalent of this phenomenon, with which we find the analogy fascinating.

[2]https://github.com/yd-kwon/POMO

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
