[Supplementary Material]

# A   Instance augmentation

## A.1   Application Ideas

**Coordinate transformation.**   TSP and CVRP experiments in this paper are following the setup where node locations are randomly sampled from the unit square, *i.e.*, $x \sim (0,1)$ and $y \sim (0,1)$. All transformations for the $\times 8$ instance augmentation used in the experiments preserve the range of $x$ and $y$, and therefore the new problem instances generated by these transformations are still valid.

For the sake of improving the inference result, however, there is no need to stick to "valid" problem instances that comply to the setup rule, as long as the (near-) optimal ordering of node sequence can be generated. Take, for example, rotation by 10 degrees with the center of rotation at $(0.5, 0.5)$. The new problem instance generated by this transformation may (or may not!) contain nodes that are outside the unit square, but this is okay. Although the policy network is trained using nodes inside the unit square only, it is reasonable to assume that it would still perform well with the nodes that are close from the boundaries. As long as the network can produce alternative result which has nonzero chance of being better and there is room in the time budget, such non-complying transformations are still worth trying during the inference stage.

Possible non-complying transformations for CO problems based on Euclidean distances are 1) rotations by arbitrary angles, 2) translations by small vectors, and 3) scaling (both bigger and smaller) by small factors. A combination of any of those, plus a flip operation, also works.

**Input ordering.**   In the AM model, the order of input data does not matter because the dot-attention mechanism does not care about the stacking order of the query vectors. But in practice, POMO can be applied to other types of neural models such as those based on recurrent structures of RNN or LSTM. For these neural nets, the feeding order of the input affects the outcome. Attention with positional encoding (as in the Transformer model) also produces different outputs with different input orderings.

The optimal solution of a CO problem should be identical regardless of the order with which the input data is given. A simple re-ordering of the input set can lead to an instance augmentation utilizable by POMO in the neural net architectures described above. This can be much more powerful than $\times 8$ coordinate transformations used in our experiments, because input re-ordering gives $N!$ number of augmentations.

## A.2   Ablation study without POMO training

In the paper, instance augmentation has been applied only on the POMO-trained networks. But instance augmentation is a general inference technique that can be adopted for other deep RL CO solvers, not just the construction-types, but also the improvement-types, too.

We have performed additional experiments that apply the $\times 8$ instance augmentation on the original AM network, trained by REINFORCE with a greedy rollout baseline. The results are given in Table 1 and 2. It is interesting that $\times 8$ augmentation (i.e., choosing the best out of 8 greedy trajectories) improves the AM result to the similar level achieved by sampling 1280 trajectories.

Table 1: Inference techniques on the AM for TSP

| Method | TSP20 | | | TSP50 | | | TSP100 | | |
|---|---|---|---|---|---|---|---|---|---|
| | Len. | Gap | Time | Len. | Gap | Time | Len. | Gap | Time |
| Concorde | 3.83 | - | (5m) | 5.69 | - | (13m) | 7.76 | - | (1h) |
| AM, greedy rollout | 3.84 | 0.19% | ($\ll$1s) | 5.76 | 1.21% | (1s) | 8.03 | 3.51% | (2s) |
| AM, 1280 sampling | 3.83 | 0.07% | (1m) | 5.71 | 0.39% | (5m) | 7.92 | 1.98% | (22m) |
| AM, $\times 8$ augment. | 3.83 | 0.01% | (1s) | 5.71 | 0.24% | (4s) | 7.89 | 1.67% | (14s) |

Table 2: Inference techniques on the AM for CVRP

| Method | CVRP20 | | | CVRP50 | | | CVRP100 | | |
|---|---|---|---|---|---|---|---|---|---|
| | Len. | Gap | Time | Len. | Gap | Time | Len. | Gap | Time |
| LKH3 | 6.12 | - | (2h) | 10.38 | - | (7h) | 15.68 | - | (12h) |
| AM, greedy rollout | 6.40 | 4.45% | ($\ll$1s) | 10.93 | 5.34% | (1s) | 16.73 | 6.72% | (3s) |
| AM, 1280 sampling | 6.24 | 1.97% | (3m) | 10.59 | 2.11% | (7m) | 16.16 | 3.09% | (30m) |
| AM, $\times$8 augment. | 6.22 | 1.63% | (2s) | 10.67 | 2.81% | (6s) | 16.35 | 4.30% | (18s) |

# B  Traveling salesman problem

## B.1  Problem setup

We need to find the shortest loop connecting all $N$ nodes, where the distance between two nodes is the Euclidean distance. The location of each node is sampled randomly from the unit square.

## B.2  Policy network

Except for the omission of start-node-selecting part, the AM model used in the POMO experiments is the same as that of Kool *et al.* [1] (which we refer to as "the original AM paper").

**Encoder**   No change is made for implementation of POMO. The encoder of the AM produces node embedding $\mathbf{h}_i$ for $1 \leq i \leq N$. We define $\bar{\mathbf{h}}$ as the mean of all node embeddings.

**Decoder**   In the original AM paper, the decoder uses a single "context node embedding," $\mathbf{h}_{(c)}$ as the input to the decoder. It is defined in Equation (4) of Kool *et al.* [1] as a concatenation

$$\mathbf{h}_{(c)} = \begin{cases} \left[ \bar{\mathbf{h}}, \ \mathbf{h}_{\pi_{t-1}}, \ \mathbf{h}_{\pi_1} \right] & t > 1 \\ \left[ \bar{\mathbf{h}}, \ \mathbf{v}^{\mathrm{l}}, \ \mathbf{v}^{\mathrm{f}} \right] & t = 1. \end{cases} \tag{1}$$

Here, $t$ is the number of iterations, and $\mathbf{h}_{\pi_t}$ is the embedding of $t$th selected node, the output of the decoder after $t$ iterations. $\mathbf{v}^{\mathrm{l}}$, $\mathbf{v}^{\mathrm{f}}$ are trainable parameters, which make $\mathbf{h}_{(c)}(t = 1)$ a trainable START token.

In POMO, we use $N$ different context node embeddings, $\mathbf{h}_{(c)}^1, \mathbf{h}_{(c)}^2, \ldots, \mathbf{h}_{(c)}^N$. Each context node embedding is given by

$$\mathbf{h}_{(c)}^i = \begin{cases} \left[ \bar{\mathbf{h}}, \ \mathbf{h}_{\pi_{t-1}}^i, \ \mathbf{h}_{\pi_1}^i \right] & t > 1 \\ \text{none} & t = 1. \end{cases} \tag{2}$$

We do not use context node embedding for $t = 1$, as POMO does not use the decoder to determine the first selected node. Instead, we simply define

$$\mathbf{h}_{\pi_1}^i = \mathbf{h}_i \qquad \text{for} \quad i = 1, 2, \ldots, N. \tag{3}$$

## B.3  Hyperparameters

Node embedding is $d_{\mathrm{h}}$-dimensional with $d_{\mathrm{h}} = 128$. The encoder has $N_{\mathrm{layer}} = 6$ attention layers, where each layer contains a multi-head attention with head number $M = 8$ and the dimensions of key, value, and query $d_{\mathrm{k}} = d_{\mathrm{v}} = d_{\mathrm{q}} = d_{\mathrm{h}}/M = 16$. A feed-forward sublayer in each attention layer has a dimension $d_{\mathrm{ff}} = 512$. This set of hyperparameters is also used for CVRP and KP.

# C  Capacitated vehicle routing problem

## C.1  Problem setup

There are $N$ customer nodes whose locations are sampled uniformly from the unit square. A customer node $i$ has a demand $\hat{\delta}_i = \delta_i/D$, where $\delta_i$ is sampled uniformly from a discrete set $\{1, 2, \ldots, 9\}$ and

$D = 30, 40, 50$ for $N = 20, 50, 100$, respectively. An additional "depot" node is created at a random location inside the unit square. A delivery vehicle with capacity 1 makes round trips starting and ending at the depot, delivering goods to customer nodes according to their demands and restocking while at the depot. We allow no split deliveries, meaning that each customer node is visited only once. The goal is to find the shortest path for the vehicle.

### C.2 Policy network

As CVRP has a fixed starting point (the depot node), POMO is applied on the second node in the path (the first customer node to visit). The original AM method feeds the depot node as the input to the policy network (*i.e.* the deport node serves as the START token), which then chooses the second node. In our POMO method, we generate $N$ different trajectories by designating all customer nodes to be the second nodes.

In TSP, all trajectories are made of the same number of nodes, which makes the parallel trajectory generation quite simple. In CVRP, the vehicle is allowed to make multiple stops at the depot. Better planned routes tend to make fewer returns to the depot, making those trajectories shorter than the others. For parallel processing of multiple trajectories, we force the vehicle with no more deliveries to stay at the depot, with a fixed probability 1, until all other trajectories are finished. This makes all trajectories to have equal lengths, simplifying parallel calculations using tensors. Note that this does not change the total travel length of the vehicle. More importantly, this does not change the learning process of the neural net as the gradient of a constant probability 1 is zero.

## D  0-1 knapsack problem

### D.1 Problem setup

We prepare a set of $N$ items, each with weight and value randomly sampled from $(0, 1)$. The knapsack has the weight capacity 12.5, 25, 25 for $N = 50, 100, 200$. The goal is to find the optimal subset of items that maximizes the total value while fit in the knapsack.

### D.2 Policy network

We reuse the neural net developed for TSP and apply it to solve KP, as both TSP and KP have an input of the same form: $N$ number of tuples $(x, y) \in \{x \in \mathbb{R}, y \in \mathbb{R} : 0 \le x \le 1, \ 0 \le y \le 1\}$. In TSP, $(x, y)$ is the $x$- and $y$-coordinate of a node, while in KP it is weight and value of an item. In KP, a "visit" to an item (a node) is interpreted as putting it into the knapsack. Inclusion of the first and the last selected items in the "context node embedding" ($\mathbf{h}_{\pi_{t-1}}$ and $\mathbf{h}_{\pi_1}$ in Equation (2)) does not seem necessary for building a good heuristic for KP, but we have taken the lazy approach and have left the model unmodified, letting the machine choose what information is relevant.

Some changes are necessary, however. When solving TSP, only visited nodes are masked from selection. When solving KP, selected items (*i.e.* visited nodes) as well as items that no longer fit inside the knapsack are masked. In the case of TSP, an episode ends when all nodes are selected. In KP, it ends when all leftover items have larger weight than the knapsack's remaining capacity. To make multiple trajectories to contain an equal number of items (for parallel processing), auxiliary items that have zero values and weights are used.

Training algorithm is modified as well. For TSP, to minimize the tour length, negative tour length is used as reward . For KP, total value of selected items (without negation) is used as reward.

## E  Our implementation of the original AM

Results of the original AM (trained by REINFORCE with a greedy rollout baseline) in our paper are slightly better than those reported in the original AM paper, for both TSP and CVRP. This improvement is mainly due to the fact that we have continued training until we observe the convergence in the training curve using more training instances.

For the full disclosure, here are a few more other changes we have made in our implementation. We update the critic network (from which the greedy rollout baseline is calculated) after each training

epoch no matter what, without the extra logic that decides whether to update the critic network or not based on its performance compared to that of the actor. We use more (6) attention layers than the original AM paper ($N_{\text{layer}} = 3$), so that the original AM and the POMO-trained AM we compare have the same structure. The batch size is fixed to 256 instances for all problems. When the training curve has converged, we apply one-time learning rate decay with a factor of 0.1 and continue training for a few more epochs.

## References

[1] Wouter Kool, Herke van Hoof, and Max Welling. Attention, learn to solve routing problems! In *International Conference on Learning Representations*, 2019.