[Reviews · NeurIPS 2020]

Review 1

Summary and Contributions: This paper proposes an approach to recover multiple optimal solutions when deep reinforcement learning is applied to solve a combinatorial optimization problem. The main idea of the approach is to start from different initial points and return multiple good solutions. The numerical experiments show that near-optimal solutions can be obtained. ** Update ** I thank the authors for their response. On one hand, the authors have responded to my general comments on "learning" mainly by giving a general overview about how important it is to solve practical combinatorial optimization problems. It is certainly important. That is why thousands of papers using exact mathematical models, heuristics, search algorithms, and so on have been published in the past. On the other hand, the authors have not responded to my comment about feasibility. Even in the rebuttal, the authors have carried on making bold claims without substantiating them. My initial assessment remains.

Strengths: The empirical results show that starting from multiple solutions and synchronizing through a shared baseline produce quite good solutions. This simple approach could be used by those researchers in the NeurIPS community, who are already trying reinforcement learning to solve combinatorial optimization problems.

Weaknesses: There are bold claims in the paper that are properly substantiated. Even the first sentence of the abstract is hard to convince those people who work in the combinatorial optimization field. In fact, obtaining better solutions to combinatorial optimization problems with reinforcement learning is more-or-less futile as there already exist extremely efficient mathematical programming methods to solve those problems. There could be one hope, if the researchers indeed manage to "learn an approach" that would reveal a policy, which could be applied to problems of various types and sizes. This paper does not give any insight about what is learned or what kind of policy is generalizable. The proposed multi-start approach is simple but not novel enough. In capacitated vehicle routing and knapsack problems, the authors did not deal with feasibility concern explicitly but opt for a "loose implementation."

Correctness: In general the empirical methodology seems correct. However, there are few points which are glossed over: - The authors claim that POMO produces multiple trajectories and each of these trajectories corresponds to an optimal solution. How do we guarantee that these are actually optimal? - The training times are not reported. Surely, these times should differ for problems with different sizes.

Clarity: The flow of the paper is okay. As I mentioned above, there are a few points which are not clear: - Lines 104-105: What do you mean by with "enough training"? Surely, trying all permutations for TSP would also reveal all the optimal solutions. - Lines 121-122: How does the shared baseline foster competition? Can you please elaborate? - Line 252-253: Why would anyone use deep neural net as a replacement for well-known and well-tested operations research methods?

Relation to Prior Work: Not exactly. The authors could have listed their contributions explicitly.

Reproducibility: Yes

Additional Feedback: Please see my comments above.


Review 2

Summary and Contributions: The paper proposes a sampling technique in reinforcement learning for combinatorial optimization problems with symmetries in the solution representation. The goal is to learn a policy that can find a number of good, but equivalent solutions. The authors test their approach on benchmarks of the traveling salesperson, capacitated vehicle routing and knapsack problem and show improvements of performance.

Strengths: The suggested sampling technique makes intuitively sense and seems to come with a significant improvement in practical performance. The results are relevant to the field of reinforcement learning for combinatorial optimization, as they may change the way we apply RL to this type of problems (which are usually non-sequential in nature).

Weaknesses: The proposed modification of the learning algorithm is a rather simple idea, not a deep theoretical contribution. Therefore, its significance may be questionable. I think however, that the experimental results may justify the importance of this contribution. --- Update --- After the author response and the discussion I would like to keep my original score. There are some concerns by the other reviewers and I cannot argue strongly for acceptance. I believe, however, that there is some value in the proposed idea and the authors may want to explore this (exploitation of formulation symmetries for learning) in more depth going forward.

Correctness: The contributions are easy to understand and I have no reason to doubt the correctness and reproducibility of the results.

Clarity: In terms of writing the paper is among the better ones. It is easy to read and the line of thought is very clear. The authors added good figures and focus on the main points of their method such that the article can be understood even with basic knowledge in reinforcement learning.

Relation to Prior Work: The differences to prior work are discussed appropriately.

Reproducibility: Yes

Additional Feedback: There are some redundant words and typos: line 32 "to" line 43 "knapsack" line 119 "a" line 208 "trajectories" line 221 "node" line 240 "the" I think in Section 5.1 it is not mentioned how many samples (N) are drawn by POMO


Review 3

Summary and Contributions: This paper aims to improve discrete optimization problems using machine learning. The proposed method, POMO, makes use of multiple starting points. Performance improvements are shown on 3 NP-hard optimization problems.

Strengths: The results seem compelling and appear to show improvements on standardized benchmarks. The method makes sense, and is clearly described. It's also a very simple method. The use of instance augmentation is quite clever.

Weaknesses: The proposed method is relatively simple which is both a good thing and a bad thing. I would argue its clearly laid out, with all (mostly) 2 main components ablated and thus fine for acceptance. That being said, I would encourage the authors to explore more ambitions research directions! It appears the proposed method acts as a heuristic way to obtain more diverse rollouts. As such, I would have expected some discussion to max entropy RL methods. If my understanding is correct, there is a rich literature that provides more principled ways of maintaining diverse rollouts. This concept of policy entropy, and the effect your method has on it could also be measured. Hyper parameters / architectural changes from past work described in the appendix complicate the story. Ideally ablations could be added either using the exact setup used in AM. It's unclear the effect.

Correctness: The discussion on baseline's for POMO to me are a bit misleading. This is somewhat of a nit though. First, the use of "traditionally" is incorrect. Earliest work (including the REINFORCE paper if I recall correctly) make use of a rolling average baseline. Newer works do use more complicated baselines, but for a reason! The baseline's accuracy directly relates to the variance of the REINFORCE estimator. If the variance reduction obtained from a more complex baseline isn't worth the compute its not worth using. The fact that you don't need a complex baseline is interesting and compelling, but going further and saying this is better feature of this algorithm doesn't feel accurate unless you demonstrate that your method somehow has lower variance gradients when using REINFORCE. In the abstract the paper states "with an extended exploration capability to find all optimal solutions." This is not correct. This method has no guarantees and does not in fact find all optimal solutions.

Clarity: Mostly the paper is clear and easy to follow. typo: 32: "We let our network to discover multiple trajectories"

Relation to Prior Work: Instance augmentation is similar to test-time augmentation performed in supervised learning. I would include a reference to this.

Reproducibility: Yes

Additional Feedback: nit: On 29, POMO is not implemented as a neural network. The policy in POMO is. Update: Thank you for your responses. After discussion with other reviewers, I intend to keep my score. The comments from the other reviewers (mostly R1) concern me and at this time I don't feel comfortable updating my score higher.


Review 4

Summary and Contributions: This paper presents POMO, a reinforcement learning (RL) algorithm leveraging multiple optima for combinatorial optimization (CO). POMO makes use of the symmetry in the CO itself and its solutions to force the network to learn multiple solution sequences by conditioning the network on the first token in the sequence, where the first token is sampled randomly from the problem. Experiments on several COs demonstrate POMO's ability to find higher quality solutions efficiently.

Strengths: The paper proposes a novel reinforcement learning (RL) algorithm for solving combinatorial optimization (CO) problems. The proposed algorithm is able to leverage the symmetries inside the problem solutions by starting from multiple designated starting points during training and inference, to force the network to learn diverse optimal solutions. The proposed approach provides an alternative methodology for dealing with symmetry or invariance in data, which should be of interest to machine learning researchers working on problems that share similar structural patterns, and people working on combinatorial optimization. To validate their approach, the authors carefully design the experiments, compare the proposed algorithm with existing CO solvers and neural-based approaches, and yield significant and convincing results. The paper is well written and easy to understand.

Weaknesses: - More ablation studies needed in the experiment section. The authors should include more ablations studies to help the audience understand the contributions of different components of the algorithm to the final performance. For each of the baselines other than AM, the authors can include the following ones, if applicable, a. baseline + multiple starting points at inference time, b. baseline + multiple starting points + instance augmentation at inference time.

Correctness: yes

Clarity: yes

Relation to Prior Work: yes

Reproducibility: Yes

Additional Feedback: - Include ablation studies for baselines other than AM as described above to help the audience understand how much POMO benefits from training and testing. - Empirically demonstrate that the algorithm is able to find multiple optimal solutions by showing one of the problem instances and POMO solutions. Edit: the authors have added more ablation studies and addressed most of my concerns.

[Author Response · NeurIPS 2020]

**Broader impacts: CO problems in industry and deep RL methods.** With POMO, our long-term primary aim is to solve hard, practical CO problems that arise in industry, manufacturing and logistics in particular. We demonstrated that POMO can solve three benchmark CO problems of different natures using the same neural net and the same training method. Human guidance during training was minimal, as we only needed to provide problem-specific scoring functions to the machine and the rest was automatic. Such adaptability and autonomy are extremely important features for industrial applications, where optimization process must be performed under various constrains and should adapt to changing environments rapidly.

From finding more efficient routing plans for goods to optimal assignments among groups of tasks and machines, majority of the problems encountered in operation research (OR) are CO problems. These problems are still dealt with hand-crafted heuristics as a common practice. In the field of computer vision and natural language processing, classical methods based on manual feature engineering by experts have now been superseded by automated deep learning algorithms. Our work signifies that such change is also possible for OR using deep-RL methods.

Traditional heuristic techniques used in OR, however, have remarkable performance. This creates a high barrier for deep RL approaches to gain meaningful attention from industry, yet. Machine learning-based heuristic solvers for CO problems that have a comparable performance to traditional rule-based solvers are extremely rare. We strongly believe that our POMO approach needs a special recognition to properly fuel the AI research efforts towards the OR practice.

**Contribution.** Contribution of our paper is three fold. First, we identify symmetries in RL methods for solving CO problems that can be leveraged to create a powerful heuristic solver. Second, we propose POMO method that rigorously takes advantage of the found symmetry: (1) Multiple trajectories are generated, each having a different but equivalent optimal solution as its goal for exploration. (2) A newly devised, low-variance baseline for policy gradient is used to update the policy. It is based on the total rewards from heterogeneous trajectories, and thus it is less vulnerable to local minima. (3) Instance augmentation technique is used to further exploit the symmetry at the inference stage. Third, we empirically demonstrate the effectiveness of the POMO method on three classic NP-hard benchmark problems with new state-of-the-art results.

**Reviewer 1.** [Weakness, Line 252, Contribution] We appreciate your valuable comments. Our answers regarding your comments are given above, and we will include them in our paper. [Correctness] In line 6, we will change the text to "... with an extended exploration techniques towards all optimal solutions." | We will add training times to our paper. [Clarity] We will change "With enough training" to "When training converges" in line 104. | With different starting points, no two trajectories can be identical. By the shared baseline, such heterogeneous trajectories are compared to each other to find the common optimal reward. Without the shared baseline, trajectories are assessed independently and the best performing trajectory is no longer guaranteed to be reinforced most. We will add more explanations to line 121.

**Reviewer 2.** [Summary, Weakness] POMO is more than a sampling technique. Behind the significant improvement on the experimental result is the new policy gradient calculation with lowered variance. Also, achievement of better trained neural net compared to the previous work based on the same model proves that our training method is highly resistant to falling into a local minimum as explained in Section 4.2. [Feedback] We will fix the typos. We will include values of N in Section 5.1 (N=50 for TSP50, N=100 for TSP100, etc.).

**Reviewer 3.** [Weakness] POMO assigns a single rollout for each optimal solution, and multiple rollouts are natural byproduct from the existence of multiple optima that indeed maximizes entropy on the first action. Relation between POMO and a max-entropy RL method is interesting, and we will add a discussion related to this topic. | The core architecture from the past work has not been changed. As for hyperparameters, we will rerun experiments and update our results to match them to avoid confusion from readers. This has marginal effect in our experience. [Correctness] We will change the word "Traditionally" to "In previous deep-RL research on CO problems." Abstract will have the change "... with an extended exploration techniques towards all optimal solutions." [Others] We will fix lines 32 and 29 as suggested, and we will include references to augmentation techniques used in supervised learning.

**Reviewer 4.** [Weakness, Feedback] Ablation studies for POMO using baselines other than attention model (AM) is an interesting topic that probably needs another paper. All the other baselines are based on "improvement type" methods (see paragraph "Construction vs. improvement") that offer no obvious way to define multiple starting points, each leading to different optimum. A separate ablation study only on instance-augmentation component is possible, however, using AM without multiple starting points, the results of which we will add to our paper. Some of the result is displayed in the table on the right; it is interesting that ×8 augmentation (i.e., choosing the best out of 8

| Method | TSP 50 | | TSP 100 | |
|---|---|---|---|---|
| | Len. | Gap | Len. | Gap |
| AM, greedy | 5.80 | 1.76% | 8.12 | 4.53% |
| AM, sampling | 5.73 | 0.52% | 7.94 | 2.26% |
| AM, ×8 aug. | 5.73 | 0.53% | 7.95 | 2.37% |

greedy trajectories) improves the AM result to the same level achieved by 1280 sampling trajectories. | We will also add graphs showing POMO's solutions for random problem instances to help visualize its empirical performance.

[Meta-Review · NeurIPS 2020]

Three reviewers support accepting the paper, one argues for rejection. From the reviews, rebuttal and discussion, the consensus seemed to be that the paper has an interesting new idea and good empirical results. The debate was around how much novelty there is, and how likely it is for the idea to be useful in the future, which are slightly more subjective concerns. I recommend acceptance, and I hope future work will show that this was a valuable stepping stone. I still recommend that the authors revise the paper according to the reviewer's suggestions, in particular in terms of not making overstated claims and giving the reader broader context.